# Big Data Analytics Using Cloud Computing Based Frameworks for Power Management Systems: Status, Constraints, and Future Recommendations

**DOI:** 10.3390/s23062952

**Published:** 2023-03-08

**Authors:** Ahmed Hadi Ali AL-Jumaili, Ravie Chandren Muniyandi, Mohammad Kamrul Hasan, Johnny Koh Siaw Paw, Mandeep Jit Singh

**Affiliations:** 1Faculty of Information Science and Technology, Universiti Kebangsaan Malaysia, Bangi 43600, Selangor, Malaysia; 2Computer Centre Department, University of Fallujah, Anbar 00964, Iraq; 3Department of Electronic & Communication Engineering, Universiti Tenaga Nasional, Km 7, Jalan Ikram-Uniten, Kajang 43009, Selangor, Malaysia; 4Department of Electrical, Electronic and System Engineering, Faculty of Engineering and Built Environment, Universiti Kebangsaan Malaysia, Bangi 43600, Selangor, Malaysia

**Keywords:** data mining, big data, cloud computing, parallel computing, power system

## Abstract

Traditional parallel computing for power management systems has prime challenges such as execution time, computational complexity, and efficiency like process time and delays in power system condition monitoring, particularly consumer power consumption, weather data, and power generation for detecting and predicting data mining in the centralized parallel processing and diagnosis. Due to these constraints, data management has become a critical research consideration and bottleneck. To cope with these constraints, cloud computing-based methodologies have been introduced for managing data efficiently in power management systems. This paper reviews the concept of cloud computing architecture that can meet the multi-level real-time requirements to improve monitoring and performance which is designed for different application scenarios for power system monitoring. Then, cloud computing solutions are discussed under the background of big data, and emerging parallel programming models such as Hadoop, Spark, and Storm are briefly described to analyze the advancement, constraints, and innovations. The key performance metrics of cloud computing applications such as core data sampling, modeling, and analyzing the competitiveness of big data was modeled by applying related hypotheses. Finally, it introduces a new design concept with cloud computing and eventually some recommendations focusing on cloud computing infrastructure, and methods for managing real-time big data in the power management system that solve the data mining challenges.

## 1. Introduction

In the early stage of the development of condition monitoring technology, most of the monitoring systems were developed for a certain type of equipment [1,2], and each system was scattered and isolated [3]. This was an information island without data sharing and interaction and was not conducive to the management and comprehensive analysis of monitoring data [4]. In addition, hardware resources such as network, computing, and storage of these monitoring systems are also difficult to share, resulting in a waste of IT facilities [5]. Therefore, an integrated management system built in the main control room has emerged, which can centrally process various monitoring data collected by different monitoring devices [6,7]. In order to integrate monitoring devices of different specifications into the centralized monitoring system, the State Grid Corporation of China issued a number of technical regulations and communication protocols, and established state monitoring centers for power transmission and transformer equipment in many provincial grid companies [8]. However, a current limitation of the monitoring device is it can only upload the processed simplified data to the monitoring center, and the data collection frequency is not high [9,10]. However, with the popularization and application of a high-speed optical fiber network and wireless transmission technology in the power industry, the future power equipment condition monitoring center will be able to receive panoramic real-time condition monitoring data from a wider area and become a data center for data integration and information sharing [11]. Therefore, the amount of collected data in the monitoring center will be staggering in the future, and the information processing capability of the existing monitoring system will be insufficient to meet the storage and processing requirements of such massive data [12,13]. Obviously, the serial processing method has long been unable to meet the processing requirements of large amounts of data [14,15]. The conventional parallel computing paradigm based on high-performance computers has always been responsible for various computing problems encountered in scientific research and engineering practice [16]. There are also many deficiencies in the state data of electrical equipment [7]. In recent years, cloud computing technology sprouted from parallel computing and has developed rapidly, and its many advantages have brought new ideas for the establishment of monitoring center computing platforms, which have attracted the attention of scholars in the power industry [2,17], as shown in Figure 1. From the research status, most of the cloud computing platforms currently designed for monitoring centers are based on a single Hadoop framework, which has certain limitations. Hadoop is good at batch processing of big data, but it cannot meet the richer computing modes in power equipment condition monitoring, such as stream computing [18]. Especially under extreme conditions (such as severe weather), the condition monitoring alarms were activated for power equipment [19].

### 1.1. Research Background

The power grid dispatch center is developing towards the promotion of online monitoring technology that integrates power regulation and control. The power equipment operating status data will be sent to the regulation center in the future. The existing Supervisory Control and Data Acquisition with power Management Systems (SCADA/EMS) are challenging for dealing with the massive amount of monitoring data [20]. In order to enable monitoring devices of different manufacturers to be connected to a unified centralized monitoring system, a state monitoring center was established for power transmission and transformation equipment in many grid companies [2]. Although web service technology reduces the difficulty of data integration, challenges remain in meeting the real-time data requirements of electric power companies [14,15].

Therefore, the current monitoring system receives skilled data processed locally in the monitoring device. For example, the monitoring device must process high-voltage electrical equipment’s partial discharge waveform signal into the number of discharges, the peak discharge volume, and the corresponding discharge phase before being uploaded [21,22,23]. Uploading “familiar data” instead of “raw data” can reduce network transmission costs and monitoring center storage costs [24]. However, a monitoring center that integrates data from different specifications of monitoring devices still faces a complex problem to in-depth state assessment and failure of the target device diagnosis [25,26].

Massive monitoring data will flood the remote power equipment monitoring center, resulting in heavy storage, processing, and analysis tasks. The condition monitoring data of power equipment has shown the main characteristics of big data: massive in volume, various in types, rapid in change, and low in value density. The monitoring data processing and analysis tasks will be transferred from distributed monitoring devices to centralized monitoring centers, which simplifies the hardware and software configuration of monitoring devices and facilitates the flexible expansion of the monitoring center infrastructure [27]. General Electric Company (GE), monitoring the operation of remote steam turbines, has shifted from the past data processing and uploading simplified data sets to the use of memory data grids and other technologies to receive and store raw sensor data [9]. The widespread application of high-speed optical fiber communication in the power industry provides solutions for massive data transmission [11].

Traditional data processing technology has encountered bottlenecks under the current explosive growth of data. It cannot meet the analysis needs of the power industry to obtain knowledge and information from massive data quickly. The power industry information is the research and application of power big data technology. It is an inevitable requirement for the development of technology and intelligence. Data mining can comprehensively use relevant algorithms to process a large amount of data and discover its hidden valuable information, which can realize the rapid conversion of data to knowledge and then value [28,29,30,31,32]. However, the current data volume is growing rapidly, and traditional algorithms based on single-node serial mining are no longer suitable for massive data requirements. As a distributed data processing platform, cloud computing can integrate many computer resources and increase technical capabilities considerably. It is more suitable for processing massive data than ordinary algorithms [14]. In addition, cloud computing models and cloud computing platforms do not have high requirements for network nodes, and regular computers can also participate in cloud computing, which reduces the complexity and cost of cloud platform construction to a certain extent [33,34,35].

In recent years, cloud computing has received more attention with a computing model that is integrated from the development of conventional computers and technological networks such as distributed computing, parallel computing, network storage, etc. [2,19,36]. The virtualization, distributed storage, and parallel computing technologies in cloud computing provide new ideas for constructing computing platforms for data centers’ power equipment condition monitoring. It is possible to integrate the existing basic computing facilities of electric power enterprises and provide reliable, stable, and powerful storage and computing capacity support which is beneficial to monitor the online power equipment over a wide range [37,38].

Monitoring and information collection improve real-time analysis and intelligent diagnosis capabilities [19,39]. With the rapid development of cloud computing technology, various high-reliability and high-scalability, big data processing systems such as Hadoop, Spark, and Storm have emerged, providing favorable tools for the centralized processing of large-scale power equipment monitoring data [40,41,42]. Although these emerging computing models all offer a unified programming interface and shield more low-level details than traditional parallel computing programming models, how can they be introduced into the data processing of the power equipment monitoring center? Combining specific professional backgrounds to solve practical problems and different high-level applications in the monitoring system is still a topic worthy of study [7,16]. The massive data characteristics in smart grid many studies are consistent with the widespread 5 V vast data with their 9 V’s Characteristics paradigm as shown in the Figure 2 [43,44,45,46].

#### Power System Condition Monitoring

Online monitoring refers to the continuous or periodic automatic collection of electrical, physical, chemical, and other state information of the monitored object with the help of monitoring devices installed on or near the monitored object without power failure. It can pass the field bus, Ethernet, wireless, and other communication methods transmit the status data to the remote monitoring system for centralized storage and processing [18,47], as shown in Figure 3. The online monitoring system uses advanced information processing and diagnosis technology to process and comprehensively analyze the status data, which can predict the remaining life of the target equipment and provide a data basis and basis for the status maintenance [48].

At the same time, testing instruments capable of infrared, ultrasonic, and other non-electricity measurements have also appeared. Since the 1990s, sensor technology, computer technology, and digital signal acquisition technology have brought online monitoring into a new stage. A multifunctional microcomputer online monitoring system has appeared, which has realized the monitoring of more equipment parameters, such as dielectric loss factor, electrical capacity, leakage current, partial discharge, etc. This monitoring system has more complete functions and improved software and hardware configuration, which can realize data processing, analysis, diagnosis, alarms, and visualization of results [49]. Integration, automation, and intelligence have become the current development direction of online monitoring technology [50]. However, most of the existing power monitoring systems are in the stage of isolated operation, and the utilization of large amounts of data obtained is low [4].

In recent years, it can only allow partial monitoring subsystems to upload simplified “familiar data” and cannot transmit sampled data of high-frequency signals. Such as the discharge signal of electrical equipment, and these data are beneficial for fault diagnosis of equipment such as transformers and GIS switches [49]. The popularization and application of high-speed optical fiber communication technology will allow the transmission of large amounts of data [50,51]. The future power equipment monitoring system will receive panoramic real-time status monitoring data from a wide range of power equipment and become data that can realize data integration and information sharing center [52]. 

Most traditional grid SCADA systems collect data through circular inquiry, which cannot collect continuous monitoring data of power equipment. It is challenging to meet real-time processing requirements and dynamic access of streaming big data for power equipment monitoring [20]. The recent emergence of cloud computing technology may bring hope to difficulties in collecting, storing, and processing big data for power equipment monitoring [53]. Ref. [19] proposed information management and computing platform architecture based on cloud computing technology for smart grid status monitoring, in which the computing layer only uses the Hadoop framework. In [54] discussed the feasibility and necessity of the power system utilizing by cloud computing, explained the features and structure goals of the power system utilizing by cloud computing center, to propose system structure design by taking simulation computing as an example to meet the needs of future smart grids for the storage, sharing, and processing of massive panoramic information. Ref. [55] proposed computing platform provides a storage and computing environment for business applications in the data center. Ref. [18], combined the characteristics of intelligent substation state monitoring data, studied the online monitoring data processing platform of substation equipment based on Hadoop, and focused on the storage and fast query methods of monitoring data in the column storage distributed database “Apache HBase”. Ref. [19] studied an intelligent diagnosis system for substation equipment based on a cloud platform, provided a variety of intelligent diagnosis methods, and realized collaborative fault diagnosis and hierarchical diagnosis through information fusion strategies.

### 1.2. Challenges

Online monitoring of intelligent power primary equipment and conventional power equipment has been dramatically developed and has become a trend [2,19,56]. Monitoring data is becoming more and more massive, and online monitoring system of power equipment faces huge technical challenges, like (real-time, rapid change, high precision, different application (various types), abnormal data brought by power data, large-scale data with complex structures, large dimensions, and massive data) [27]. Research work can be roughly divided into two kinds of detection methods based on traditional detection methods and detection methods based on data mining [14]. Traditional power load abnormal detection methods are generally based on human experience, state estimation, load curve, similarity and load change rate [57]. Due to the low efficiency, strong subjectivity, and many human factors of this method. This method is suitable just for abnormal data with large abrupt changes, and cannot detect abnormal data with unobvious changes [58]. With the continuous development of artificial intelligence and cluster analysis theory, the speed of data mining technology in the detection of abnormal power load data is also increasing [59]. where commonly can be roughly divided into two categories: firstly, the abnormal value detection method with supervised learning, which selects a part of the power system load data as the training sample and then uses the corresponding algorithm to make the selected sample data and the expected output meet the corresponding requirements based on support vector machines, detection methods based on artificial neural networks, and detection methods based on decision trees [60]. secondly, the unsupervised learning power load abnormal data detection method does not need to select part of the historical power load data as training samples [61]. This type of abnormal data detection method usually includes density analysis, cluster analysis and so on. The distance-based outlier detection algorithm belongs to the unsupervised algorithm. This method is easy to understand and explain, and it is one of the most representative methods in density-based outlier detection algorithms, and it has better results when dealing with medium and high-dimensional data [61]. The density-based abnormal load data detection method still has the problem that some parameters are specified by subjective factors such as human experience, which reduces the accuracy of abnormal data detection [62]. The abnormal value detection method based on the clustering algorithm can more accurately distinguish the normal data and abnormal data in the electric load data set according to the characteristics of the electric load data. 

At present, cloud computing technology has achieved good results in the fields of medical data storage, traffic data real-time analysis, and weather data analysis [37,63]. The cost of cloud computing is low, and there are no complicated requirements for the servers in the cluster establishment [64]. With the help of the large scale and fast calculation speed of cloud computing, the management and analysis of power monitoring data can be realized more effectively through integration with traditional data mining techniques [33,65]. In summary, the power load and abnormal data detection method based on data mining and cluster analysis has become a research hotspot in recent years because of its ability to dig deeper into the changing law of the load curve and effectively detect abnormal load data, and a certain result has been achieved [12,66]. In practice, the types of power load data are complex and diverse, and the contradiction between the increasing scale of power load data and the low efficiency of data mining algorithms has gradually become prominent. Each clustering algorithm has defects such as difficulty in optimizing the determination of initial parameters and high sensitivity [67]. Therefore, the related algorithms for the parallel detection of power load data need to be further studied. All the above explanation summarizes in Figure 4.

At present, the application research of cloud computing-based big data technology in the power industry is still in the exploratory stage [30]. Presently, most cloud computing platforms designed for monitoring centers are based on a single Hadoop framework and have certain limitations storing data before centralized processing that lead to processing time delay is too long [29], which cannot meet the requirements of online monitoring data [28,42]. Meet the data flow quickly and efficiently will become a trend of information processing in the future [12,68]. Despite several studies on velocity, volume, and variety, a full and efficient solution is currently unavailable on the market; the most popular method is to use a database management system (DBMS) that may be incompatible with older systems [69]. The rapid processing of monitoring data and the problem of abnormal detection in real-time are becoming increasingly prominent [13]. The introduction of cloud computing technology into the power field has essential research significance, but the application in the power industry is still in the exploratory stage, and more in-depth research is needed to put it into power production [70]. Cloud computing technology has attracted much attention with high performance, but how to use cloud computing technology for large-scale real-time data processing has not been studied [71]. Additionally, multisource heterogeneous urban sensor access and data management technologies provide strong support for intelligent perception and scientific management at the city scale and can accelerate the construction of smart cities or digital twin cities with virtual reality features [72,73,74]. The difficulty of the processes has frequently increased with the single computing resources insufficient because of the complicated operation of the combined prediction model and meet in real-time for intelligent power systems [7,39,75]. The challenge to analyzing and studying the electricity consumption massive data lead to necessitate the development of clustering analysis algorithms combined with cloud computing [69]. In most of the studies of cloud computing, used as a local controlling system [76,77]. Therefore, there was a lack of studies that used cloud computing to benefit from the ability to share the systems that control the process more widely. Several studies proposed cloud computing products, like artificial intelligence (AI) [78], and the internet of things (IoT) [5,79,80,81] in their models, however, all of the studies were focusing only on the smart vehicles’ batteries [77,78,82,83,84]. Therefor this research focuses on the problem of centralized parallel processing and diagnosis of power system condition data based on cloud computing and big data technology because difficult to meet the real-time requirements of the power system and construction of a safe, stable, cost-effective, green and environmentally friendly smart grid. The challenges of power management and monitoring have many problems; this work is proposed to target three different Problems Figure 5.

Meet the multiple real-time requirements of power system condition monitoring;The weakness of traditional data mining based on single-node serial mining;The insufficient algorithm that combines data mining and computing technology to deal with massive data.

Finally, cloud computing could be a great addition to an intelligent power system that is aimed at solving the challenge of massive data from large-scale areas of a smart power system. Contributions to this literature evaluation take into account the aforementioned issues. To give an overview of big data analytics using cloud computing frameworks for power management systems at the moment, we propose the first universally applicable framework for parallel optimization in power systems, which researchers can use to systematically describe their parallelization studies and place them in the landscape of parallel optimization without regard to the application domain, problem addressed, methodology parallelized, or technology used. The proposed approach, in particular, incorporates both algorithmic design and computational implementation challenges of parallel optimization, which are often dealt with separately in the literature. Second, we use the integrative framework to consolidate earlier research in the field of power systems on parallel optimization.

Furthermore, there are also difficulties managing information and data due to the millions of intelligent meters that need to be managed effectively. Cloud computing could offer a more affordable option for data analytics and storage., as shown in Figure 6, and Table 1 [85].

### 1.3. Novelty

Recently, Ref. [7] introduced a framework that contains an EMS stored on the cloud computing service as a combination of monitoring different power sources as well as managing the charging and discharging process. This model is useful for handling the best-optimized system and for controlling the switches the power hub before the need to deal with the data that comes from the power system. Therefore, this study focuses on solving this problem by use of the first step of the optimum system that will be Big Data Analytics Using Cloud Computing Based Frameworks for Power Management Systems through an overview of big data analytics using current cloud computing frameworks for power management systems, and we propose the first universally applicable framework for parallel optimization in power systems, which researchers can use to systematically describe their parallelization studies and place them in the landscape of parallel optimization without regard to the application domain, problem addressed, methodology parallelized, or technology used. The proposed approach, in particular, incorporates both algorithmic design and computational implementation challenges of parallel optimization, which are often dealt with separately in the literature. Second, we use the integrative framework to consolidate earlier research in the field of power systems on parallel optimization. Cloud computing could be a great addition to any system aiming for an optimal solution, especially intelligent power systems that solve the challenge of massive data from large-scale areas of a smart power system. All these processes and data will be saved and controlled by a cloud computing framework using a cloudsim.

### 1.4. Organizing of Paper

The flowchart is demonstrates the layout and organization of the manuscript in graphical form as follows [91]:



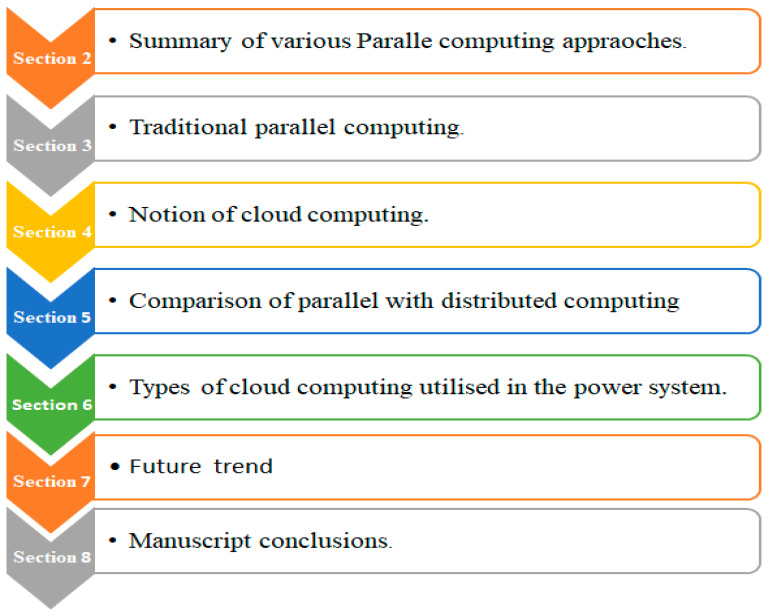



## 2. Parallel Computing

### 2.1. Concept of Parallel Computing

Parallel computing is a computing model relative to serial computing [92]. In this mode, multiple computing processes are performed simultaneously, which is different from the conventional computing method that is performed sequentially [93]. As shown in Figure 7, users write a series of computer instructions for specific problems to form tasks. In serial computing mode, these instructions can only be executed one by one by a single processing element (PE). Decomposition into multiple subtask instruction sets that allow simultaneous solving is mapped to multiple PEs for execution. Where a PE refers to a single-core central processing unit (CPU) or a core of a multi-core CPU, and is the basic hardware unit for executing instructions [94].

### 2.2. Classification of Parallel Computing Technology

There are many classification methods for parallel computing technology, and the following only introduces the classification methods related to this article [95].

#### Flynn Classification

The most classic Flynn classification is classification from the most basic instructions and data processing methods [96,97]. Figure 8 shows the four different types after classification.

(1)SISD is a traditional serial computing method. Early computers fell into this category in a certain clock cycle, only one instruction is executed and only one data stream is processed;(2)SIMD is uses one instruction to process multiple data streams simultaneously in a certain clock cycle. Current single-core computers also fall into this category and are widely used in the fields of digital signal processing, image processing, and multimedia information processing;(3)MISD is uses multiple instruction streams to process a single data stream. Currently, it is only a theoretical model and has no application examples;(4)MIMD are currently the most popular. Multicore processors fall into this category which can execute multiple instruction streams on multiple different data streams at the same time.

### 2.3. Classification by Computational Characteristics of Applications

(1) Data-intensive applications: Such applications are faced with a huge amount of data with relatively simple calculations [98]. Usually, parallel computing is performed by dividing the data, which can also be called data parallelism [26,99].

(2) Computation-intensive applications: The amount of data processed by this type of application is not large, but the calculation is very complex [100]. Generally, this method of decomposing tasks is used for parallel computing, which can also be called task parallelism [101].

(3) Mixed-intensive applications: The amount of data and computation can use either data parallelism or task parallelism to solve the problem, or a combination of the two parallel methods can be used to solve the problem [37]. Often, partitioning data is easier to handle than breaking down tasks [102].

## 3. Shortcomings of Traditional Parallel Computing

### 3.1. Computational Complexity Issues

Real-world optimization issues in a various applications fields are often NP-hard, and even the development of (meta) heuristic optimization algorithms might necessitate significant computer resources [103]. The rising availability of powerful computer capabilities will be used to solve complex optimization issues with parallel algorithms in numerous industries, including finance, logistics, production, and design [70]. A lack of unifying frameworks for parallel optimization across techniques, application sectors, and challenges adds to the heterogeneity [37,38]. The concerns of a novel integrative framework for parallel computational optimization across optimization problems, techniques, and application domains. The framework combines algorithmic design and parallel optimization computational implementation viewpoints [104]. Unsurprisingly, the application of parallel optimization has been hesitant because (i) parallelizing algorithms is challenging in general from both the algorithmic and the computational perspective, and (ii) a viable alternative to parallelizing algorithms has been the exploitation of ongoing increases of clock speed of single CPUs of modern microprocessors [105]. 

Parallelization initiatives are now considerably more crucial than they were in the past due to progress. Fortunately, the need for parallelization has been realized, and parallel computing resources are now more widely available. The phenomenon: The rapid development of parallel hardware architectures and infrastructures, including multi-core CPUs and GPUs, local high-speed networks, and large data storage, as well as libraries and software frameworks for parallel programming, is the fundamental cause of this [106]. The study discovered a large number of published reviews on parallel optimization for specific issues, approaches, uses, fields of study, and technological advancements. The majority of the assessments we discovered center on parallel optimization with respect to specific approaches. Metaheuristics have dominated methodological literature evaluations, as indicated in Table 2. Several points that demand for a new literature review have been presented in earlier reviews. In the first place, we were unable to locate reviews of the most recent research on the parallelization of both precise and (meta)heuristic approaches that had been published in the years 2008–2017. Second, the categories that were employed to define and organize earlier literature were different. Aside from this heterogeneity, there aren’t any frameworks that can be used to describe parallel optimization across techniques, application areas, and issues [77,78]. Finally, this has produced a fragmented overall view of what has been accomplished and what needs to be done in parallel optimization in operations research (OR). As a side effect, the heterogeneity with which parallelization studies in OR have been described in terms of algorithmic parallelization, computational parallelization, and parallelization performance is high. This is advantageous from a diversity perspective, but it also raises problems such as heterogeneity makes it often time-consuming and, in some cases, impossible for readers to identify the aforementioned parallelization characteristics of a study, to classify the study.

### 3.2. Multi-Source Heterogeneous Problem

Mostly in the domain of computers, heterogeneity typically refers to various instruction-set architectures (ISA), also called computer architecture, a machine that carries out the commands specified by that ISA, such as CPU [120]. The amount of heterogeneity in contemporary computing systems is progressively rising, with many new processors now featuring integrated logic for interacting with other devices (SATA, PCI, Ethernet, USB, RFID, radios, UARTs, and memory controllers), as well as programmable functional units and hardware accelerators (GPUs, cryptography co-processors, programmable network processors, A/V encoders/decoders, etc. [121]. When a system uses the same ISA but has a heterogeneous CPU topology, the speed of the cores varies [5]. In contrast to normal homogeneous systems, heterogeneous computing systems come with different issues [122]. All of the problems associated with homogeneous parallel processing systems are brought up by the presence of many processing units, and the degree of heterogeneity in the system can bring about non-uniformity in system development, programming techniques, and overall system capabilities [123]. Three categories of heterogeneity can be present: the first is instruction-set architectures (ISA): Binary incompatibility could result from differing instruction set designs in compute elements. The second is application binary interface (ABI): Different memory interpretations may be possible for compute elements [124]. This depends on the architecture and compiler being used and may include bendiness, calling convention, and memory layout. The third is application programming interface (API): It’s possible that not all compute parts will have access to all OS and library services at once [125].

In terms of interconnecting, the compute pieces may be interconnected in a variety of ways. While some parts of a heterogeneous system may be cache-coherent, maintaining consistency and coherency may be explicitly required for other parts of the system [126]. A heterogeneous system may include CPUs with equal architecture in terms of performance, but with subtle changes in microarchitecture that affect both performance and power consumption [127]. Performance predictability issues, particularly when dealing with mixed workloads, can occasionally be caused by asymmetries in capabilities combined with opaque programming models and operating system abstractions [128,129]. Regarding Data Segmentation While data splitting on homogeneous platforms is frequently easy, it has been demonstrated that the task is NP-Complete for the general heterogeneous situation [130]. It has been demonstrated that there are ideal partitioning’s for small numbers of partitions that completely balance the load and reduce communication volume [131].

### 3.3. Data-Intensive Challenges

The parallel computing facility resources are usually a single high-performance computer or high-performance computing cluster, whether they are a shared storage architecture or a distributed storage architecture or they are still logically reflected as a single high-performance computer for parallel computing users [100,132].

The computing resources and storage resources are relatively isolated physically, and the computing unit accesses data on the storage unit through a data bus or a high-speed network. For compute-intensive applications with small amounts of data, this does not pose a problem [133]. When faced with increasingly prominent data-intensive application requirements, frequent read and write accesses between computing units and storage units will become the performance bottleneck of the entire parallel system. This problem is especially true in a shared storage architecture where data is centrally stored [134]. In order to reduce the amount of communication data between processes, each worker has a local backup of all the data to be analyzed, and only the metadata of the data is sent when assigning tasks [135]. However, this precondition is not easily satisfied in clusters of distributed storage architectures, so the parallel scheme is only suitable for multi-core or many-core computers when solving data-intensive problems [136].

### 3.4. Scalability

The high-performance computer (HPC) clusters consist of high-performance hardware and are very expensive [137]. The cluster expansion generally adopts a vertical expansion method, which improves the computing performance of the cluster by replacing CPUs, expanding memory, and adding disks. However, vertical scaling is very limited, it is easy to reach the limit and the upgrade is expensive. The scalability can be achieved by horizontal expansion, such as a cluster of workstations (COW), but there are still other problems in traditional parallel computing based on this hardware [138].

### 3.5. Usability

Although the existing traditional parallel computing programming models such as Message Passing Interface (MPI) and OpenMP have been encapsulated at the bottom, and that data storage management, data division, task allocation and scheduling, data synchronization and communication, fault tolerance, and many other technical details need to be handled by users themselves, which is still very cumbersome [139]. Users are entangled in many underlying technical details while considering the application problem itself, which makes parallel programming not easy [105].

## 4. Cloud Computing 

### 4.1. Concept of Cloud Computing

The concept of cloud computing, is defined by the US National Institute of Standards and Technology (NIST): as a model that can achieve convenience for obtaining the required resources (including networks, servers, storage, applications, and services) through network access on demand, and the required resources can be quickly provided or released, with little management effort or little interaction with service providers interaction [140,141]. Cloud computing is generally considered to have the following characteristics:(1)Virtualization: Virtualization is the core technology of cloud computing, and many other features that depend on it. The application of virtualization technology can integrate heterogeneous computing resources to form a resource pool for users to access [142].(2)Service-oriented: Cloud computing provides three levels of services, namely Infrastructure as a Service (IaaS), Platform as a Service (PaaS), and Software as a Service (SaaS). IaaS is the lowest-level service that directly provides compute, memory, and networking equipment. Users have the greatest degree of freedom and can build their own platforms and software. PaaS is one level higher than IaaS, providing a ready-made cloud platform, saving the work of developing the platform. SaaS provides more convenient services; users can directly use the provided software without any development [143];(3)Elasticity and scalability: The cloud scale can be easily expanded without affecting the cloud services currently provided externally. Resources in the cloud are infinitely desirable to users and can be automatically provisioned and reclaimed quickly on demand [144];(4)Reliable and universal: Cloud computing technology provides a variety of fault-tolerant mechanisms to ensure high reliability of services [145]. Data is placed with multiple copies to prevent data loss due to hardware failure [146]. Compute services that were stopped due to hardware failures can still continue elsewhere through virtual machine migration. Virtualization makes cloud computing resources transparent to users and supports applications in different industries at the same time [147];(5)Economies of scale: The cloud computing platform does not have high requirements for hardware facilities, and a large number of idle ordinary computers can be integrated into the resource pool through virtualization [33]. For users, it saves hardware costs and daily management costs of self-built platforms [57]. For cloud service providers, the versatility of cloud computing has greatly improved the utilization of resources, and the scale has significantly increased economic benefits [148].

### 4.2. Cloud Computing Environment

A smart grid is a heterogeneous and complex environment containing different kinds of devices, networks, systems, and data. IEC 61970 and IEC 61850 were discovered via a study on the principal Smart Grid standards and are open-source platforms based on cloud computing. IEC 61970 specifies the application program interface (API), while IEC 61850 specifies the abstract communication services interface (ACSI) [149].

Compatibility with the IEC61970 and IEC61850 standards for cloud computing technologies such as Hadoop, Spark, and Storm involves careful analysis of existing systems, translation of data, adoption of standards-compliant data formats, integration of systems, and thorough testing to ensure that the systems are functioning as intended. Because these two standards defined data models and the interface separately, the models are not uniform, and seamless communication between the substation and the control center is not possible. Whereas IEC 61970 defines the power of an information model and is widely used in enterprise integration. IEC 61850 is restricted to data exchange within substation equipment. Research revealed that Hadoop might pool idle power system resources and provide “super-computing capability” for the smart grid’s data integration platform. The grid dispatch automation system’s support platform and application software should be upgraded in accordance with the component interface specification (CIS) and Common Information Models (CIM) standards. The data is integrated using the IEC61970 standard and connected to the smart grid via a platform for data sharing [150]. The main user of these standards will not be a person, but a computer and they have to be machine-readable. At the same time, they are very complex documents involving thousands of different items. The full series of IEC 61850 Standards is now available as a global package. They are issued with the available associated code components. The series includes no less than 35 documents, dealing with substation automation, DER integration or cyber security, to name but a few [151]. With the development of cloud computing technology and the needs of big data processing, a number of new computing models and programming models have emerged, providing users with a basic platform for parallel programming in the cloud environment, and shielding users as much as possible from the bottom layer. The details are presented to the user through a higher-level abstract interface [152].

#### 4.2.1. Hadoop Technology

Hadoop is a massive data processing system open sourced by the Apache Software Foundation. It includes many components for storage or processing such as the Hadoop Distributed File System (HDFS) and the MapReduce parallel computing model [153].

HDFS has the characteristics of distributed storage, high concurrent access, high fault tolerance, simple consistency and provides a reliable storage environment for parallel computing models [154]. It adopts a master-slave architecture and is built on a physical cluster connected by multiple computers through a network. The bottom layer is the local file system of the operating system. Its architecture is shown in Figure 9, shows HDFS includes a master node NameNode, a backup master node SecondaryNameNode and a set of DataNode slave nodes. The NameNode is responsible for managing the HDFS namespace, saving all metadata, and responding to client access requests. SecondaryNameNode is used to solve the single point of failure problem of Hadoop. The DataNode is responsible for the actual storage and management of redundant data blocks of files, and the data block files are actually stored on the local file system of each node [155].

The basic idea of MapReduce is basically the same as that of traditional parallel computing models such as MPI, which is to “divide and conquer” a large amount of data. The system provides two simpler interfaces, Map and Reduce which automatically completes many underlying functions such as task division and scheduling, communication, load balancing, and failure recovery [156]. Figure 10, shows the MapReduce programming mode [140]. The input file is divided into several slices (InputSplit) according to a specific format, and converted into <key, value> key-value pairs, which are input to Mapper for calculation, and the intermediate results are key–value pairs. The collection is aggregated by the Reducer after a shuffle phase, and the result is saved to HDFS. Hadoop MapReduce is mainly oriented to the batch mode of massive static data, and its real-time performance is not high [141]. The collection is aggregated by the Reducer after a shuffle phase and the result is saved to HDFS. Hadoop MapReduce is mainly oriented to the batch mode of massive static data, and its real-time performance is not high [142]. Therefore, it too quickly processes and analyze the massive historical data accumulated in the condition monitoring of power equipment, hoping to discover valuable knowledge from it [143,144].

#### 4.2.2. Spark Technology

Spark is a Hadoop MapReduce-like general-purpose parallel computing framework that appeared in 2012 [157]. The difference is that it puts data (including some intermediate data) in memory for calculation, avoiding a large amount of disk I/O caused by frequent reading and writing of HDFS during the calculation process. Therefore, spark is suitable for iterative and interactive computing scenarios, and even in general application scenarios which is more efficient than Hadoop MapReduce [158]. Spark’s in-memory computing features from its core abstraction, Resilient Distributed Dataset (RDD) [159]. RDDs are read-only, fault-tolerant, distributed computing, partitionable, coarse-grained transformations, and in-memory storage. Each partition of the new RDD generated during Spark’s calculation process has a dependency relationship with the partition of its parent RDD due to the calculation, which is called lineage. The lost RDD partition can be regenerated from the ancestor RDD by tracing the lineage, so as to implement fault tolerance. At the same time, Spark divides the entire computing process into multiple stages according to the different dependencies between RDDs, each stage generates a job, and creates tasks in units of RDD partitions and distributes them in multiple stages. Figure 11, shows the process in parallel on two computing nodes, and the Spark computing model [160]. The computing model is richer and more flexible than the single MapReduce model of Hadoop, and is compatible with various data sources such as HDFS, HBase, and Hive [161]. At present, Spark has been widely used in Internet companies such as Amazon, Yahoo, and Taobao [162], and it is still in the research stage in the power industry, and research still needs to be carried out in combination with typical application scenarios [163]. Based on Spark, is incapable of how the complex signal processing algorithm can perform fast calculation when the amount of data is large, which makes up for the application scenarios that Hadoop MapReduce [164].

#### 4.2.3. Storm Technology

Storm is primarily geared towards real-time analytics on large-scale, uninterrupted streams of data, unlike Hadoop which focuses on batch processing of massive amounts of data. Although Spark Streaming can also achieve the function of stream computing by decomposing batch jobs, its latency is longer than Storm [165]. Storm also adopts a master-slave architecture and uses ZooKeeper to coordinate the entire cluster. Where the master node is called the control node and runs the Nimbus daemon, which is responsible for publishing topology programs, distributing tasks and monitoring cluster status [166]. The slave node is called a worker node, running the Supervisor daemon, which is responsible for accepting the assigned tasks and starting the Java Virtual Machine (JVM) process worker to execute that shows in Figure 12a [167,168]. Figure 12b shows the topology where Spout is the data entry of the topology, connecting to an external data source and converting the data into tuples that are sent to Bolt. The processing logic for tuples is encapsulated in Bolts, and after the processing is completed, tuples can be transmitted to subsequent Bolts [169]. The Spout and Bolt components are linked by a stream grouping strategy and can be configured as multiple instances to achieve parallel processing. Each instance will eventually form a task to be scheduled for execution [170].

## 5. Comparison of Parallel Computing with Cloud Computing

Cloud computing is the fusion of many technologies, including parallel computing technology, so it is not equivalent to parallel computing, and its connotation richer [36,171]. The parallel computing usually refers to a specially designed parallel computer, while cloud computing combines multiple ordinary computers to achieve the purpose of improving computing performance [172,173]. In a broad sense, the computing technology it adopts also belongs to parallel computing. The technical point of parallel computing only focuses on computing and ignores data storage [174]. This is because traditional parallel programming models such as MPI are designed for high-performance computing, and small amount of data [174]. Cloud computing includes storage and computing, and the two cooperate with each other. For example, in the Hadoop system, data is stored in a distributed manner, and then the calculation is moved to the location of the data for execution, because mobile computing is more efficient than moving data [175]. From the perspective of applicable fields, parallel computing is suitable for the scientific computing field with high-performance computing requirements [2]. It is oriented to computing-intensive applications and requires users to have high professional quality in order to be able to deal with many low-level details [105]. The cloud computing provides services to users through three different levels, which is easier for users to use. The cloud computing is a key technology for big data processing and is suitable for data-intensive applications, but the large system management and maintenance costs make it not good at computing-intensive applications with a small amount of data [25]. To sum up, parallel computing and cloud computing technologies have a wide range of applications in various fields including the power industry [139,176]. The two are complementary rather than mutually exclusive, and each has different application scenarios [177].

### Distributed Cloud Computing and Parallel Computing

A new method to connect data and applications supplied from several places is distributed cloud computing. A shared resource geographically dispersed among several users or systems is referred to as distributed, according to Table 2. The ability to execute several jobs simultaneously is a key characteristic of cloud computing, which also aims to reduce CPU consumption, cut down on switching times, reduce waiting times for data processing, increase server throughput, and enhance data processing and communication speed. Another feature makes using any cloud application to communicate with users in various places simple for users. The final crucial component is enhancing server performance since communication performance is crucial.

After reviewing each reference included in Table 3, it was determined that [178] was the finest work on distributed cloud computing since it covered the most features.

The three different methods of parallel processing are distributed, shared, and hybrid memory systems. A distributed parallel computing focus identified certain key criteria, as indicated in Table 4. The important feature mentioned in more than one resource is improving performance using the load balancing technique through distributing the process and making a balance between servers for processing the jobs and improving the performance of our distributed system. Another feature is minimizing resource costs because when we divide the load among servers, we can minimize the resource cost such as CPU, memory, and storage. Since it is preferable to employ a system for handling user requests with a low response time, all of the references apply this principle by recommending a method for distributed parallel computing based on that factor. After looking through the references in this work, it was determined that Ref. [185] was superior since it provides a wide range of characteristics, such as load balancing, enhancing system performance, and lowering reaction time and resource cost.

## 6. The Application Basis of Cloud Computing in a Power System

In the early days of the Internet, the cost of hardware was relatively high. With the continuous development of information and network technology, the data generated and processed by the Internet has grown exponentially. In order to cope with these changes, the investment in hardware equipment has to be increased input cost [38]. However, although the hardware investment cost is high, the scalability of the system is very poor, the information transmission efficiency is low, and the performance difference between devices is also the processing effect is not good [171]. The theoretical basis of cloud computing is to virtualize equipment and services, and then interconnect several distributed nodes on the cloud platform to realize the superposition and coupling of computing resources [191]. In order to better complete the processing of massive data the distribute a large number of processing tasks to virtual nodes in the resource pool [29,166]. Cloud is a very broad concept with many types, and it is usually classified according to different service objects and service types [192].
(1)Public cloudAs the name suggests, it is a cloud service that is open to the public. It is large-scale, low-cost, and the most popular cloud service for the public. The most typical application is Amazon Web Services (“AWS”). The app provides a complete set of infrastructure and cloud solutions to customers around the world. AWS provides users with a complete set of cloud computing services, which can help enterprises reduce IT investment costs and maintenance costs and easily migrate to the cloud [193].(2)Private cloudIt is a cloud that does not provide services publicly and is used within a group or organization. Provide private cloud services to internal users. Because they cannot be used publicly, most firewalls are set up [194]. The typical representative of private cloud is the Blue Cloud plan launched by IBM. Blue Cloud is based on open standards and open source software powered by IBM software, systems technologies and services [195]. The Blue Cloud developed by more than 200 IBM researchers around the world, will help clients quickly and easily explore cloud computing infrastructure for extreme-scale computing [196].(3)Hybrid cloudThat is, the combination of public cloud and private cloud is between private and public, such as Amazon’s virtual private cloud (VPC) [54]. A VPC is a dynamically provisioned pool of public cloud computing resources that requires the use of encryption protocols, tunneling protocols, and other security procedures to transfer data between private enterprises and cloud service providers [197,198,199,200]. The services provided by each layer are as follows:(1)**Application layer**The application layer provides users with various application software and services required by a friendly user interface [201]. The application layer directly faces customer needs and provides enterprise customers with enterprise applications such as enterprise resource planning (ERP) and customer relationship management (CRM) [202], and office automation (OA) [203].(2)**Platform layer**The platform layer provides services for users who can use the platform to realize the value they want to achieve [204].(3)**Infrastructure layer**This layer provides infrastructure-level services, that is, the establishment of the cloud computing platform infrastructure is directly open to users, so that they can use the powerful storage and computing capabilities of cloud computing. Users can directly store files and run calculations in the cloud, and also the infrastructure can be allocated independently, which is equivalent to the user having a scalable computer with large storage space and supercomputing performance through the terminal [205].



In general, the advantage of cloud computing is that it can integrate various resources without special requirements for these computing resources and does not require a specific computer with powerful performance [33]. Users can easily obtain cloud services through ordinary terminals and utilize the capabilities of supercomputers. In the same way, an intelligent cloud system can also be built in the power system. Users can monitor and track the power system directly on the mobile terminal to improve the convenience of the power system [206,207,208].

At present, the smart grid construction the structure diagram shown in Figure 13. Compared with the previous computing model, cloud computing has made many qualitative breakthroughs. Its scale is large, but it has strong reliability [209,210]. The expansion type, which integrates these features together makes it have unparalleled advantages [211]. Google is the initiator of cloud computing ideas. Later, on the basis of Google’s previous work, developed the significant Hadoop open-source cloud computing platform. The key to Hadoop is to include two major systems: the distributed file system HDFS and the distributed computing framework [212,213].

## 7. Future Trend

Cloud-based tools and technologies for microgrid control are a set of software and hardware solutions that are designed to monitor and manage the power usage and distribution of microgrids. The architecture of cloud computing provides robust and efficient management of data, computing resources, and applications, with features like fault isolation cloud computing, which refers to the ability of the system to isolate a failure or malfunction in one part of the system so that it does not affect the entire system. Furthermore, system self-healing refers to the ability of the cloud computing system to automatically detect and recover from failures without human intervention. Where in a cloud computing environment, micro-grid control can be achieved by using cloud-based tools to monitor and control the power usage and distribution of the micro-grid. These systems can provide real-time data and insights into the power usage of individual devices and systems, allowing for more efficient and effective management of the microgrid. Local data interaction can be achieved through the use of APIs (Application Programming Interfaces) and other communication protocols that allow for seamless integration between the cloud and local systems. APIs can be used to access cloud resources and services, as well as to exchange data between the cloud and local systems. This allows for more efficient and effective use of resources and provides a convenient way to access cloud services from local systems.

In the future, power system cloud computing will have broad applications, which can be described as “the future can be expected” as shown in Figure 14. Its effective application can provide a large number of high value-added services inside and outside the industry and has high practical value for the profitability and control cost level of power companies [214].

The subsystem model has long-term benefits for national defense, military, and medical care. The application value in the power grid alone is immeasurable.

First, it will help grid companies to carry out grid operation and maintenance monitoring and improve response sensitivity [215]. Use the data collected from the power system to monitor, control, or adjust the power generation, load, and fault status in the network, and respond accordingly when there is an error or an upgrade in the power grid [209,216].Secondly, it will help grid companies conduct special analysis on equipment maintenance, operation and maintenance, improve system reliability, power supply qualification rate, reduce costs, and reduce power outages [217]. In the field of power grid maintenance, operation and maintenance, through the selection of key indicators of power equipment from the three aspects of safety, benefit, and cost, analysis of the mutual influence of “safety”, “benefit” and “cost” in maintenance management, coordination of the three these factors are comprehensively optimized, and at the same time, real-time online monitoring of the maintenance indicators of power grid enterprises is realized, providing guidance and services for the company’s maintenance strategy formulation [218,219,220].

Some points that need improvement due to the limitations of previous work are personal ability and data source:➢The data capacity is still not large enough, and the data dimension change interval is small [221]; the electricity data comes from residential electricity consumption and does not include electricity consumption data in other areas such as industrial electricity [222];➢The coverage of parallel algorithm design is relatively narrow, and the application range in power system data processing is not wide enough. However, with the increasing informatization of the power system and the continuous quantification of power data, the application scope of data mining technology continues to expand. Parallel algorithms can be designed in more aspects to enhance the data processing effect and carry out all around power system production and dispatching [223,224];➢To design a comprehensive cloud computing platform architecture based on cloud computing and big data processing technology, which provides a reference plan for the construction of the computing platform of the power equipment monitoring center [225,226];➢To improve the real-time response speed of online monitoring data of power equipment, a real-time processing framework for streaming data based on Storm is needed, and an incremental variable prediction model classification method [227,228,229].

## 8. Conclusions

This review article briefly expounds on the basic concepts and programming models of parallel computing and analyzes the shortcomings of traditional parallel computing in the face of current big data scenarios with a specific application example. Then, cloud computing solutions are introduced for the field of power equipment condition monitoring under the background of big data, the concepts and characteristics of cloud computing are introduced, and emerging parallel programming models, such as Hadoop, Spark, and Storm, are briefly described. Finally, for different application scenarios in the field of power equipment condition monitoring, a comprehensive cloud computing platform architecture is designed to meet the multi-level real-time requirements by drawing on the experience of the Internet field.

There is a need to improve the parallel detection algorithm for power load anomaly detection efficiency. The detection of the abnormal value is only the first step in the analysis of the abnormal value of the power. After the abnormal value is detected, the cause of the abnormal data can be quickly analyzed, such as economic reasons, regular maintenance, equipment failure, illegal electricity stealing, temperature, or sudden environmental changes. Forming a power load abnormal detection response model to provide guiding opinions for demand-side power supply is one direction of future research. For power load data prediction, the further parallelization of the power load data prediction algorithm is another key direction of the next work. As an important part of power data analysis, research on power load anomaly detection and prediction methods in the cloud computing environment still has huge research opportunities waiting for us to study.

## Figures and Tables

**Figure 1 sensors-23-02952-f001:**
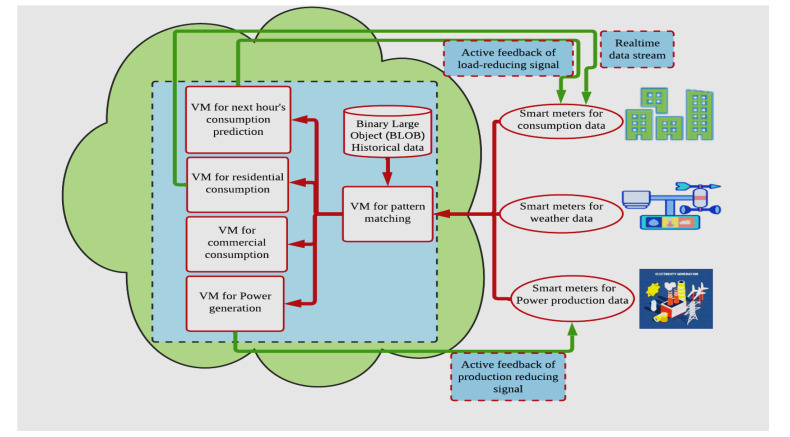
Efficient data processing for the power system.

**Figure 2 sensors-23-02952-f002:**
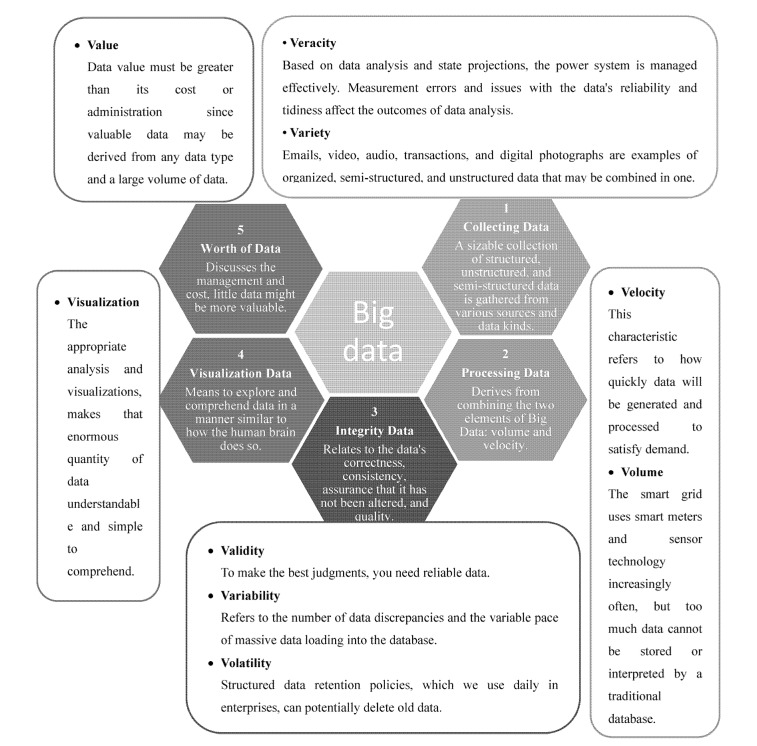
Five categories (collecting data, processing data, integrity data, visualization data, and worth of data) of big data with their 9 Vs characteristics.

**Figure 3 sensors-23-02952-f003:**
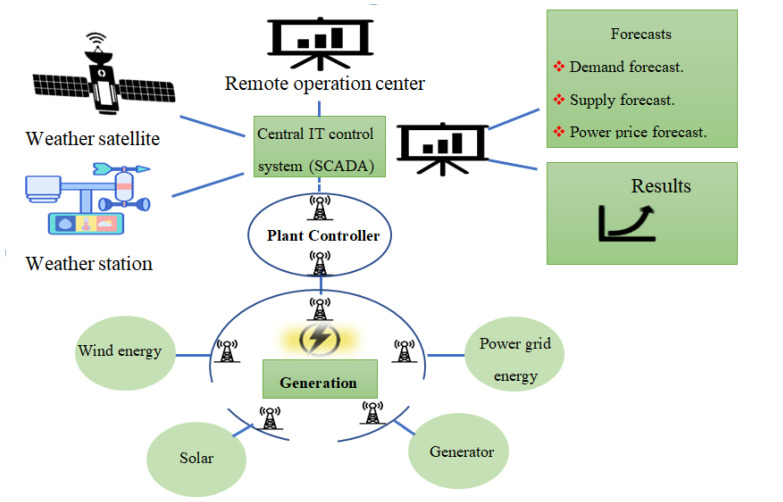
Power system controller monitoring.

**Figure 4 sensors-23-02952-f004:**
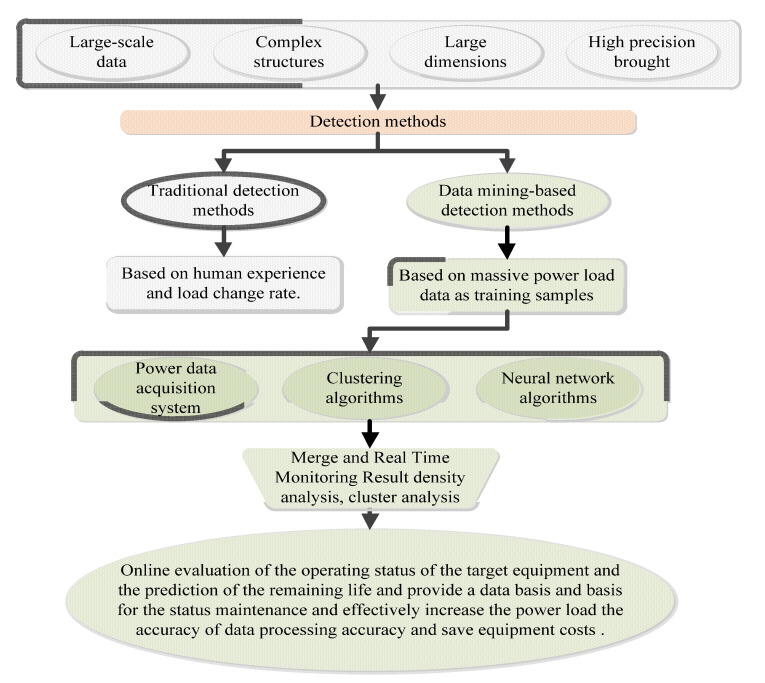
Techniques power system detection.

**Figure 5 sensors-23-02952-f005:**
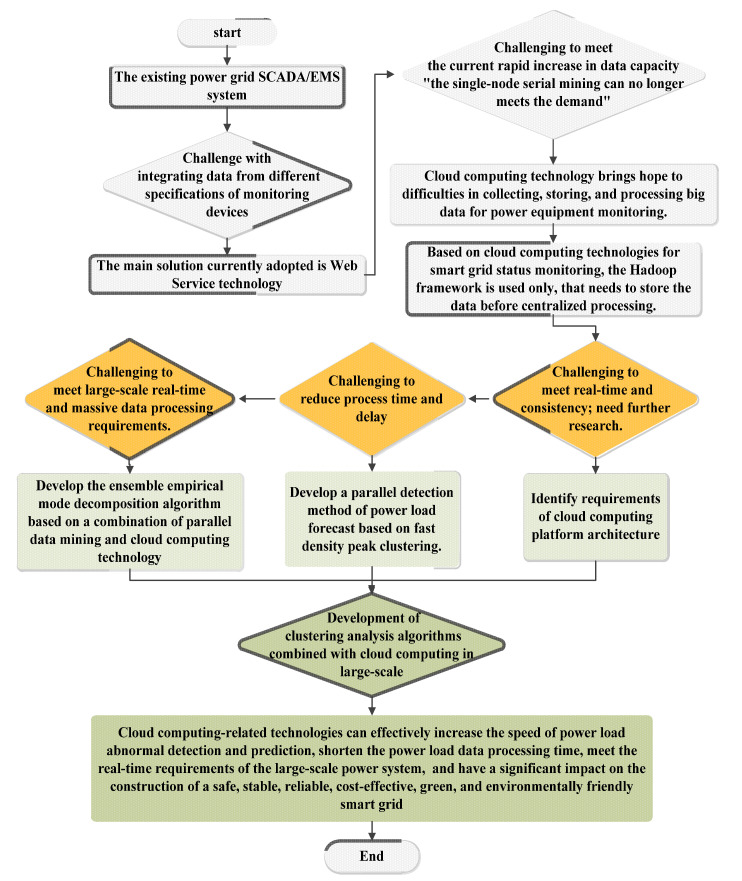
General flowchart for challenges and solutions the proposed system.

**Figure 6 sensors-23-02952-f006:**
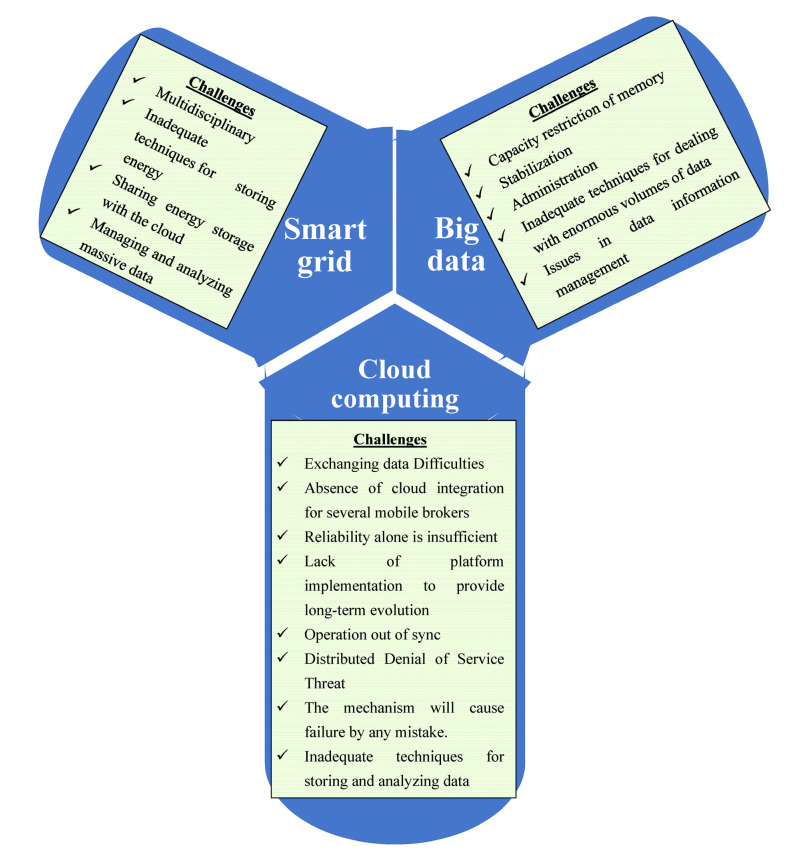
Key characteristics of large data in smart grids.

**Figure 7 sensors-23-02952-f007:**
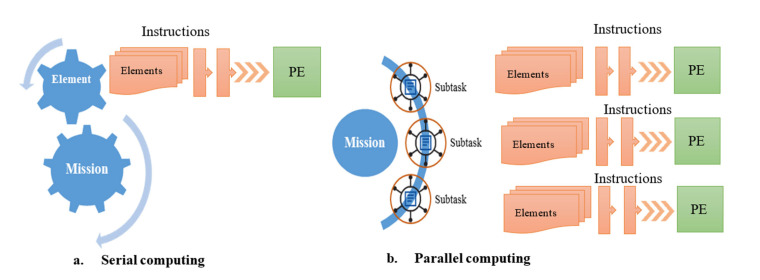
Sequential computing and parallel computing.

**Figure 8 sensors-23-02952-f008:**
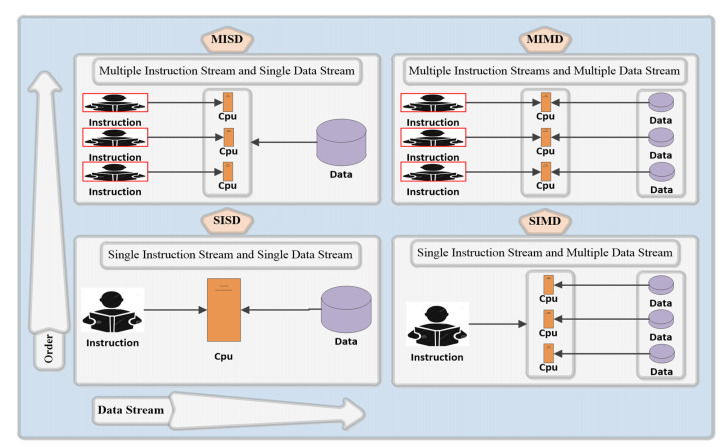
Classification by Flynn for computer systems.

**Figure 9 sensors-23-02952-f009:**
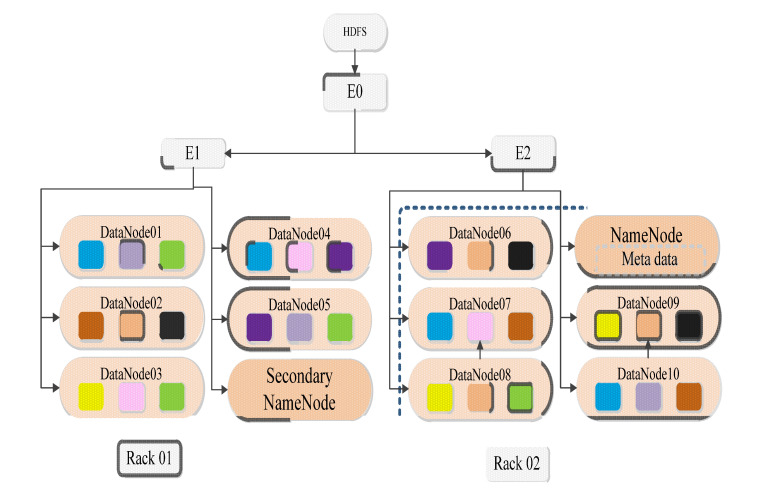
Architecture of HDFS.

**Figure 10 sensors-23-02952-f010:**
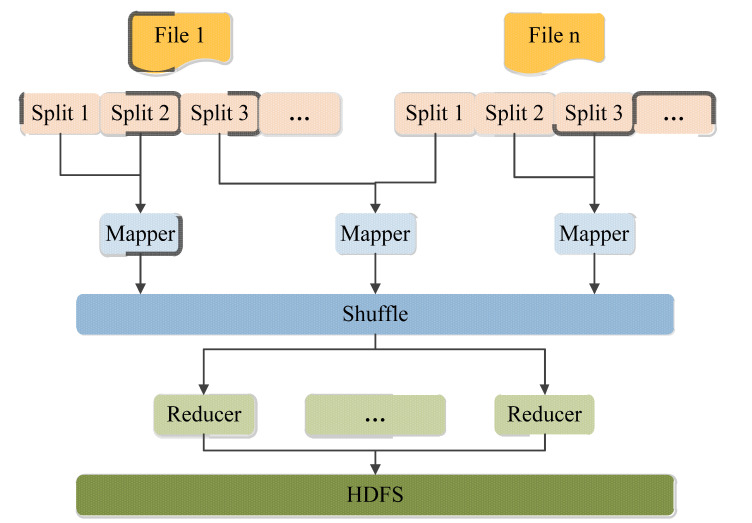
MapReduce programming model.

**Figure 11 sensors-23-02952-f011:**
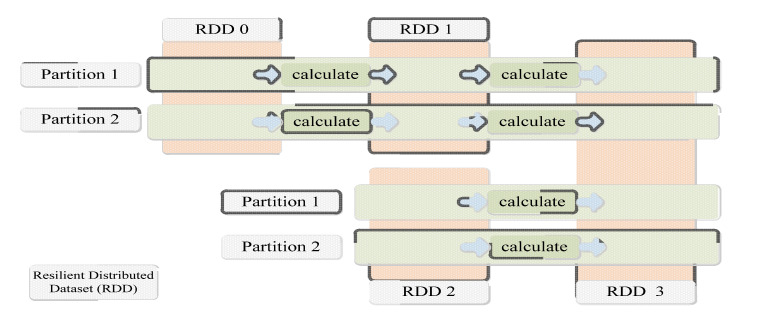
The computing model based on resilient distributed dataset (RDD).

**Figure 12 sensors-23-02952-f012:**
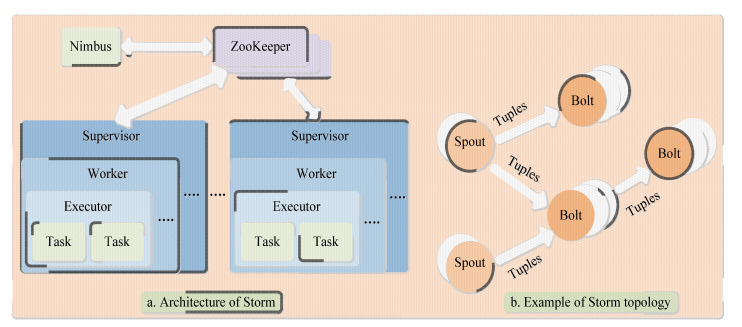
Architecture and topology of storm.

**Figure 13 sensors-23-02952-f013:**
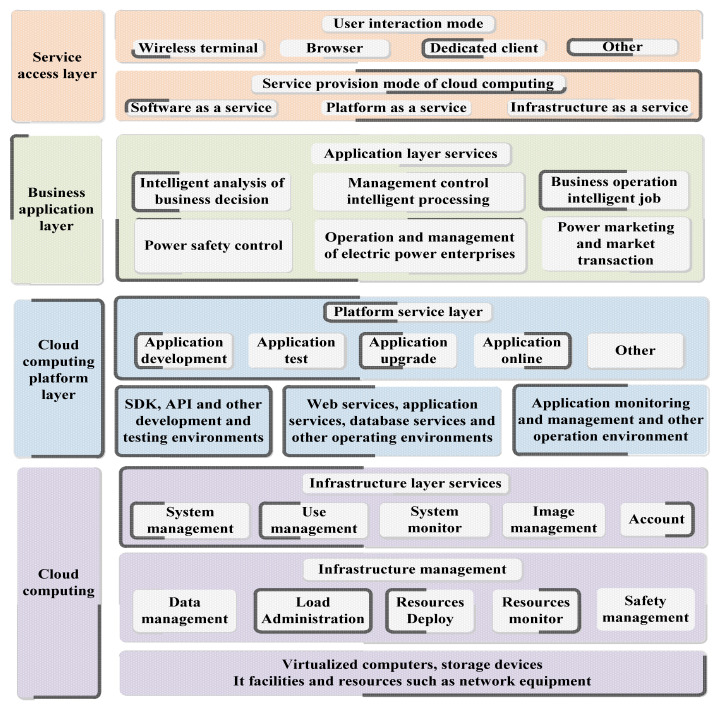
Structure of grid intelligent information platform.

**Figure 14 sensors-23-02952-f014:**
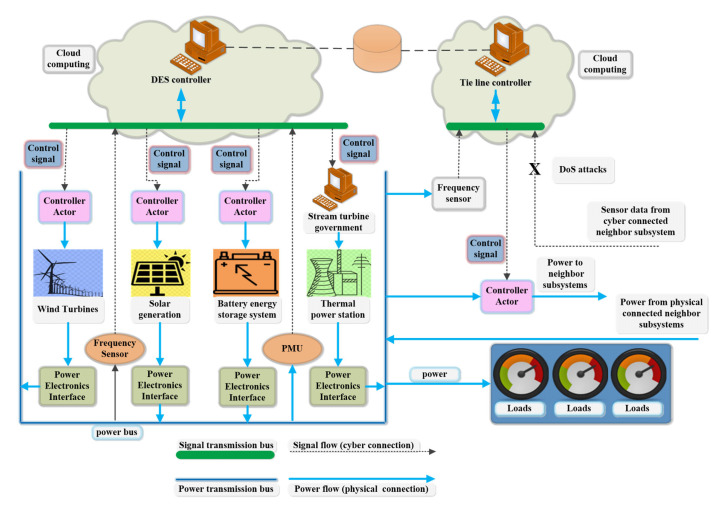
Schematic diagram of a subsystem.

**Table 1 sensors-23-02952-t001:** The difficulties managing information and data due to the millions of intelligent meters data that need to be managed effectively through cloud computing.

Ref.	Existing Challenging	Proposed Solution	Challenging Proposed Solutions and Future Work
[20]	The problem with concurrent power transmission networks is the uneven temporal distribution and the growing number of fault occurrences that cause power outages or interruptions.	The suggested model incorporates and explicitly assesses seldom occurring environmental components, faults, and periods with fewer fault events, which improves the forecast performance of power transmission fault events.	challenging to deal with the massive amount of monitoring data
[24]	The problem of conventional clustering algorithms for Big Data Analytics.	Parallel algorithms of k-means and canopy are implemented using the Hadoop environment and Mahout to solve the problem of conventional clustering algorithms.	Process locally after storage data.
[14]	Problems with traditional data mining that is generated in single-node chain mining.	This work uses single-node serial mining to tackle the classic data mining problem in power systems. It has vast storage and processing capacities, and accuracy 87%.	Did not use cloud computing so it was hard to meet real-time and large scale
[2]	The limitations of centralised administration based on LAN design prevent broad-area monitoring and the resolution it’s issues.	This study describes cloud-based power grid-wide-area monitoring architecture for parallel computing and big data mining to give intelligent grid decisions.	This paper’s flaw is a lack of data exchange during processing.
[31]	Considering real-time application, the smart grid still needs to advance in terms of efficiency, power management, dependability, and value.	Using cloud computing architecture from any location and at any time, design remote real-time monitoring of substation power data in a safe, efficient, and effective manner.	The weakness of this work the power flow in the grid is continuously monitored using PLC and Energy Meter, it doesn’t use cloud computing applications.
[19]	Large-scale data processing and analysis methods in a real-time panoramic grid are a challenge for smart grids.	This paper use data mining and integrated information technology platform to present a smart grid building a large multi-level data storage system to extract valuable knowledge to support grid scheduling decisions.	Dealing with redundant data and noise in data mining results remains a barrier for technology. It is also uncertain if the current cloud platform will get real-time smart grid monitoring data.
[70]	As smart grids spread, terminal devices like cutting-edge sensors and smart metres tend wide access to distribution networks, providing major challenges to the information perception, analysis, and processing capacities of the distribution automation system.	This paper aims at guiding to preserve CPU and memory resources and increase resource utilisation. through presents a configuration technique for computing resources for the microservice-based edge computing apparatus in the smart distribution transformer region.	The lack is the trade-off methods between robustness and economy in computing resource configuration problems and apply the achievement of this work to investigate the computing resource scheduling problem of the cloud-edge collaborative system in the smart grids.
[71]	It has become very difficult to process big amounts of real-time data in research and applications, and it hasn’t been researched how to employ cloud computing technology for large-scale real-time data processing.	This research focuses on the big data processing architecture of the cloud computing platform. It creates a large data processing calculation mode and establishes the overall real-time big data processing architecture that acts as the foundation for the RTDP (Real-Time Data Processing)	The RTDP is a tough project, and many issues still need to be researched further: Choosing the most effective technique for calculating future design performance; Real-time data processing hardware must be implemented equally.
[72]	The huge challenge of integrating and exchanging vast sensor information resources that differ widely in hardware design, connection protocols, formatting, conversational skills, sampling rate, and data accuracy.	This paper provides a deeper understanding of the needs, platforms, most current technical developments, and open research problems of urban sensor applications for academics and leaders in the IoT and smart cities sectors.	Relational databases usually struggle with scalability, availability, and concurrent reading and writing, especially for big data handling in wireless sensor networks. As IoT and sensor technology continues to progress, cloud computing will be used.
[86]	The ability to detect and analyse anomalies for huge data in real-time is a tough problem due use conventional detection methods of data processing.	An anomaly detection model based on Hadoop distributed processing method, cloud computing and MapReduce monitoring framework is presented using machine learning.	The challenge to Meeting the real-time and large scale
[16]	Data from networks and smart cities is increasing and it is becoming huge so it need to big data analysis (BDA)	BDA generated in the smart city (IoT) to turn the smart city toward safety, efficient data processing, and good governance.	The flaw is the system created for the study only offers offline batch analysis and prediction functions.
[87]	Smart grids (SGs) are utilizing massive data for operations and services.	Information and communication technologies (ICTs) play an important role, particularly in the computing model, which governs how data analytics in SG may be carried out.	The design of EC systems, EC-appropriate algorithms, resource management in the EC environment, and even hardware accelerations might all be improved.
[88]	Increasing renewable energy sources making the power system more complex.	This study focuses on using ICT data in smart grid decision-making to ensure systems are secure and reliably operate.	The SCADA issues caused by ICT integration continue to exist like interdependency analysis, and decision-making.
[89]	There are challenges to controlling MGs in a logical and coordinated way	In this study, control objectives are categorized in line to the hierarchical control layers in MGs, and the development approaches given by MGSC/EMS are summarized.	the challenging issue is the uncertainty about power production related to weather, load calculation times and response time brings more challenges to MGSC/EMS.
[21]	The challenge of extracting data value through the statistical analysis of an immense amount of data generated by cyber-physical systems.	The goal of this paper was not to give the solutions, but rather to name the problems. A major challenge is the changing nature of the technical systems	software-based devices change frequently due to bug fixing and software updates. Therefore, the data we collected is after time only partially valid.
[90]	The challenge of clustering techniques in Big Data context.	Provide a thorough analysis of the Big Data clustering problems and highlight the benefits of the key methods.	Data are too big, dynamic, and complex. Traditional data handling struggle to collect, store, and analyse data.
[28]	The execution of the Hadoop cluster when processing a high number of tiny files is the true problem businesses face. The solutions are restricted to NameNode memory	Some novel strategies have been put forth, such as combining tiny heterogeneous files in various formats in a quasirandom manner, which resolves the memory issue by drastically reducing the amount of metadata.	Hadoop cannot satisfy real-time demands because it stores data before processing.
[29]	Big Data poses difficulties for Digital Earth in terms of data mining, processing, and storage. Transforming big data’s volume, velocity, and diversity into values is the main challenge.	Cloud computing provides fundamental support to address the challenges with shared computing resources including computing, storage, networking and analysis, that fostered Big Data advancements.	It is extremely difficult to achieve in real-time processing.
[30]	Large data environments lack capabilities like support for massive data, high performance, high reliability, scalability, and high resource.	This paper studied features of popular NoSQL and NewSQL databases for unified storage management and quick data access.	It is extremely difficult to achieve in real-time processing.
[69]	Big data is currently the most difficult organisational problem due to the rapid generation of new data every second. Systems cannot be compatible with typical DBMS solutions.	In order to address diversity in greater detail, this article discusses current problems, possibilities, trends, and difficulties associated with big data. We’ll talk about an effective fix for the huge data variety issue.	It is extremely difficult to achieve in real-time processing.

**Table 2 sensors-23-02952-t002:** Several parallel optimization methods and technologies use Metaheuristics.

Algorithm	Reference
Tabu search (TS)	[107]
Simulated annealing (SA)	[108]
Variable neighbourhood search (VNS)	[109]
Greedy Randomized Adaptive Search Procedures	[110]
Swarm intelligence algorithms	[111]
Particle swarm optimization algorithms	[112]
Genetic algorithms (GAs)	[113]
Ant colony optimization algorithms	[114]
Scatter search	[115]
Several reviews have covered sets of Metaheuristics	[116]
Hybrid Metaheuristics	[117]
General-purpose computation on graphics processing units (GPC-GPU), in particular, are noteworthy parallelization approaches	[118,119]

**Table 3 sensors-23-02952-t003:** Distributed Cloud Computing.

Feature	[178]	[179]	[180]	[181]	[182]	[183]	[184]
Reduction in CPU use	✓						
Reduce multiple-process tasks	✓	✓		✓		✓	✓
Reduce waiting times.	✓						
enhanced use of resources	✓						
Enhance server efficiency	✓						✓
Increasing server performance	✓						
Balance of loads			✓				
performance in terms of costs			✓				
lessen the demand on the memory		✓		✓			
Create a cloud architecture program				✓	✓		✓
enhance inter-humans communication					✓		
Increasing safety					✓		
Increasing effectiveness and creating a system expand					✓		
Increase the scope of cloud computing						✓	
Comparison of the benefits and drawbacks of MPI, oprnMPI, and MapReduce						✓	

**Table 4 sensors-23-02952-t004:** Distributed Parallel Processing.

**Features**	[181]	[182]	[186]	[187]	[188]	[189]	[185]	[190]
Utilize load balancing to increase performance	✓	✓	✓	✓		✓	✓	✓
Requesting each node’s status				✓				
developed a reduce reaction time-based algorithm			✓		✓	✓		
Decrease requests on resources that are available					✓			
Minimize server-to-server interaction and processing					✓			
Take every resource’s load into account						✓		
Optimize CPU throughput							✓	
Reduce productivity							✓	
Reduce reaction time.							✓	
Reduce long waits							✓	
lessen the cost of resources			✓				✓	
Ensure error tolerance and QoS							✓	
Effective implementation of parallelism	✓							
Improving the way jobs are arranged			✓					
Improve allocation of resources			✓					
Faster performance with better outcomes			✓					

## Data Availability

Not applicable.

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
