# Peer review of "Big Data Analytics Using Cloud Computing Based Frameworks for Power Management Systems: Status, Constraints, and Future Recommendations"

_sensors, 2023, doi:10.3390/s23062952_

Round 1
Reviewer 1 Report
| The article has the good quality to be published in this journal. Only the following source should be used in it.
Power efficient virtual machine placement in cloud data centers with a discrete and chaotic hybrid optimization algorithm S Gharehpasha, M Masdari, A Jafarian Cluster Computing 24, 1293-1315 |
||
| Virtual machine placement in cloud data centers using a hybrid multi-verse optimization algorithm S Gharehpasha, M Masdari, A Jafarian Artificial Intelligence Review 54, 2221-2257 |
Author Response
Sensors Journal MDPI
ISSN: 1424-8220
Submitted to the section: Internet of Things, Advanced Technologies in Sensor Networks and Internet of Things,
Title: Big Data Analytics Using Cloud Computing-Based Frameworks for power Management Systems: Status, Constraints, and Future Recommendations.
Reviewer-1# Comments and Suggestions for Authors.
The article has the good quality to be published in this journal. Only the following source should be used in it.
Concern # 1:
Power efficient virtual machine placement in cloud data centers with a discrete and chaotic hybrid optimization algorithm
S Gharehpasha, M Masdari, A Jafarian Cluster Computing 24, 1293-1315. Virtual machine placement in cloud data centers using a hybrid multi-verse optimization algorithm S Gharehpasha, M Masdari, A Jafarian Artificial Intelligence Review 54, 2221-2257.
Author's response:
- We would like to thank the reviewer for this comment. We agree with the reviewer that it is a very important aspect. We have addressed and considered the research, furthermore, added references to current articles.
Authors’ action:
- The two citations addressed in (17 and 65), Page No. (2 and 7) and Line No. (14 and 3 ) of the manuscript of the article.
Reviewer 2 Report
This paper studied the “Big Data Analytics Using Cloud Computing Based Frameworks
for power Management Systems: Status, Constraints, and Future Recommendations". The quality of the article is Suitable. Minor revision should be done for this version of the paper as follows:
* Some figures like figure 2 have low quality.
*The language usage throughout this paper need to be improved, the author should do some proofreading on it.
Author Response
Sensors Journal MDPI
ISSN: 1424-8220
Submitted to section: Internet of Things, Advanced Technologies in Sensor Networks and Internet of Things,
Title: Big Data Analytics Using Cloud Computing Based Frameworks for power Management Systems: Status, Constraints, and Future Recommendations.
Reviewer-2# Comments and Suggestions for Authors.
This paper studied the “Big Data Analytics Using Cloud Computing Based Frameworks
for power Management Systems: Status, Constraints, and Future Recommendations". The quality of the article is Suitable. Minor revision should be done for this version of the paper as follows:
Concern # 1:
Some figures like figure 2 have low quality.
Author's response:
- We appreciate the reviewer's feedback, and we thank him for it. The figurines have been updated and redrawn with high-quality results. Figure 2 in particular has been redrawn.
Authors’ action:
- The figure 1 is addressed in Page No. (2) and Line No. (19 to 20) of the manuscript of the article.
- The figure 2 is addressed in Page No. (4) and Line No. (9 to 21) of the manuscript of the article.
Concern # 2:
The language usage throughout this paper need to be improved, the author should do some proofreading on it.
Author's response:
- We would like to thank the reviewer for their insightful feedback. We agree with the reviewer that this is an essential element. Indeed, following a thorough review, we have revised and updated the English grammar mistakes in the article's manuscript using professional and native English speakers.
Authors’ action:
- The revised text was on all pages of the manuscript of the article.
Reviewer 3 Report
This paper studied Cloud Computing Based Frameworks for power Management Systems, which mainly introduces the basic concepts and research status. It’s written well on the conclusion of current research situations, which is comprehensive. However, there are still some problems need to be solved.
1) It would be better to take a deeper consideration of the actual demand of the electric power system. Different electric power system services have different requirements for real-time and reliability of data processing and analysis. For instance, power system security and stability analysis require fast response, and power system market transactions have different scales such as day-ahead market and day-to-day market. Accordingly, it is necessary to design a cloud computing architecture based on the actual business requirements of different data in the power system.
2) It’s not very necessary that all data need to adopt the architecture of cloud computing and analyze it in the cloud. For example, fault isolation, system self-healing, micro-grid control and other functions, local data interaction can be achieved. With the analysis of these situations, this paper would be more comprehensive and accurate.
3) This paper is mainly a brief introduction to the existing cloud computing technology, such as Hadoop, Spark, Storm and heterogeneous system integration, but doesn’t explain how to design the technical scheme according to the different actual business requirements of the power system, and also doesn’t mention how to compatible or transform with the existing IEC61970 and IEC61850 standards.
Author Response
Sensors Journal MDPI
ISSN: 1424-8220
Submitted to section: Internet of Things, Advanced Technologies in Sensor Networks and Internet of Things,
Title: Big Data Analytics Using Cloud Computing Based Frameworks for power Management Systems: Status, Constraints, and Future Recommendations.
Reviewer-3# Comments and Suggestions for Authors.
This paper studied Cloud Computing Based Frameworks for power Management Systems, which mainly introduces the basic concepts and research status. It’s written well on the conclusion of current research situations, which is comprehensive. However, there are still some problems need to be solved.
Concern # 1:
It would be better to take a deeper consideration of the actual demand of the electric power system. Different electric power system services have different requirements for real-time and reliability of data processing and analysis. For instance, power system security and stability analysis require fast response, and power system market transactions have different scales such as day-ahead market and day-to-day market. Accordingly, it is necessary to design a cloud computing architecture based on the actual business requirements of different data in the power system.
Author's response:
- We would like to thank the reviewer for this comment. We agree with the reviewer that it is a very important aspect. Indeed, after a careful examination, we have currently revised and updated the font mistake in the manuscript of the article.
Authors’ action:
- The revised text was addressed in section 6: on Page No. (25), lines No. (4 to 24), Page No. (26), lines No. (1 to 36), and Page No. (27), lines No. (1 to 2) in the manuscript of the article.
Concern # 2:
It’s not very necessary that all data need to adopt the architecture of cloud computing and analyse it in the cloud. For example, fault isolation, system self-healing, micro-grid control and other functions, local data interaction can be achieved. With the analysis of these situations, this paper would be more comprehensive and accurate.
Author's response:
- We would like to thank the reviewer for this comment. We agree with the reviewer that it is a very important aspect. When talking about how micro-grid control and other functions, local data interaction can be achieved, With the analysis of these situations. Furthermore, the future, power system cloud computing has a broad prospect, which can be described as the future can be expected.
Authors’ action:
- The revised text was addressed in section 7: on Page No. (27), lines No. (6 to 16), Page No. (28), lines No. (1 to 22), and Page No. (29), lines No. (1 to 20) in the manuscript of the article.
Concern # 3:
This paper is mainly a brief introduction to the existing cloud computing technology, such as Hadoop, Spark, Storm and heterogeneous system integration, but doesn’t explain how to design the technical scheme according to the different actual business requirements of the power system, and also doesn’t mention how to compatible or transform with the existing IEC61970 and IEC61850 standards.
Author's response:
- We would like to thank the reviewer for this comment. We have addressed the details of the design the technical scheme according to the different actual business requirements of the power system.
Authors’ action:
- The revised text was addressed in section 1.3 Challenges on Page No. (6), Lines No. (21 to 43), Page No. (7), Lines No. (1 to 18), Page No. (8), Lines No. (1 to 41), Page No. (9), Lines No. (1 to 5), and Page No. (10), Lines No. (1 to 5) in manuscript of the article.
- The revised text was addressed in section 4.2 Cloud computing environment on Page No. (19), Lines No. (24 to 43) in manuscript of the article.

Round 2
Reviewer 3 Report
- I think the current version can meet the requirements and have no further comment.